# Phytophenol Dimerization Reaction: From Basic Rules to Diastereoselectivity and Beyond

**DOI:** 10.3390/molecules27154842

**Published:** 2022-07-28

**Authors:** Shuqin Liu, Xican Li, Ban Chen, Xiaojian Ouyang, Yulu Xie, Dongfeng Chen

**Affiliations:** 1School of Chinese Herbal Medicine, Guangzhou University of Chinese Medicine, Guangzhou Higher Education Mega Center, Waihuang East Road No. 232, Guangzhou 510006, China; 20201110616@stu.gzucm.edu.cn (S.L.); imchenban@foxmail.com (B.C.); oyxiaojian55@163.com (X.O.); xieyulu1900@163.com (Y.X.); 2School of Basic Medical Science, Guangzhou University of Chinese Medicine, Guangzhou Higher Education Mega Center, Waihuang East Road No. 232, Guangzhou 510006, China

**Keywords:** radical coupling, diastereoselective, stereoselectivity, phenolic, phytophenol

## Abstract

Phytophenol dimerization, which is a radical-mediated coupling reaction, plays a critical role in many fields, including lignin biosynthesis. To understand the reaction, 2,2-diphenyl-1-picrylhydrazyl radical was used to initiate a series of phytophenol dimerization reactions in methanol. The products were identified using ultra-performance liquid chromatography coupled with electrospray ionization quadrupole time-of-flight tandem mass spectrometry (UHPLC-ESI-Q-TOF-MS/MS) analysis in situ. The identified products mainly included biphenols, magnolol, honokiol, gingerol 6,6′-dimers, 3,6-dimethoxylcatechol β,β′ dimer, euphorbetin, bis-eugenol, dehydrodiisoeugenol, *trans-ε*-viniferin, (+) pinoresinol, and (−) pinoresinol. Structure–function relationship analysis allowed four basic rules to be defined: *meta*-excluded, C–C bonding domination, *ortho*-diOH co-activation, and exocyclic C=C involvement. The exocyclic C=C involvement, however, required conjugation with the phenolic core and the *para*-site of the -OH group, to yield a furan-fused dimer with two chiral centers. Computational chemistry indicated that the entire process was completed via a radical coupling reaction and an intramolecular conjugate addition reaction. Similar results were also found for the horseradish peroxidase (HRP)-catalyzed coniferyl alcohol dimerization, which produced (+) and (−) pinoresinols (but no (−) epipinoresinol), suggesting that the HRP-catalyzed process was essentially an exocyclic C=C-involved phytophenol dimerization reaction. The reaction was highly diastereoselective. This was attributed to the intramolecular reaction, which prohibited *Re*-attack. The four basic rules and diastereoselectivity can explain and even predict the main products in various chemical and biological events, especially oxidase-catalyzed lignin cyclization.

## 1. Introduction

The phytophenol dimerization reaction is ubiquitous in nature and has been recognized for over 100 years [1]. Fungi, algae, and insects use the reaction to construct fruiting bodies, cell walls, and cuticles. Particularly, plant cells utilize the phytophenol dimerization reaction toward the biosynthesis of lignin and lignan, and to construct cellulose, the most abundant natural product found on land [2,3]. Essentially, phytophenol dimerization reaction is radical-mediated coupling reaction [4]. Until now, the basic rules of the phytophenol dimerization reaction have remained poorly understood. This has limited the clarification of lignin biosynthesis. The ~60% interunit linkage and most polycyclization of lignan, for example, have not been unlocked until now [5]. Therefore, there is an urgent need to better understand the basic rules and stereochemistry of phytophenol dimerization reaction [6].

However, the understanding is limited by the complexity and variability. Usually, the phytophenol dimerization reaction involves four structural factors, namely the coupling position (e.g., *ortho*, *meta*, and *para*), bonding type (e.g., C–C and C–O), phenolic -OH content, and exocyclic C=C bond. If an exocyclic C=C bond involves in the phytophenol dimerization reaction, it may produce a new chiral carbon center. This situation is much more complicated than other dimerization reactions (e.g., the Norrish reaction).

Undoubtedly, the essential approach to establishing the basic rules is to access key chemical information from the phytophenol dimerization products obtained. However, these products are unstable and variable because they are easily oxidized by air due to the presence of one or more phenolic -OH group [7,8]. A conventional phytochemical approach is impractical for product identification. 

To overcome the limitations of product identification, cutting-edge ultra-performance liquid chromatography coupled with electrospray ionization quadrupole time-of-flight tandem mass spectrometry (UHPLC-ESI-Q-TOF-MS/MS) has been used in this study. In general, MS data (such as *m*/*z* values) are insufficient toward identifying the structure of a product. However, when MS data are compared with those of authentic standards or detailed documented data, the product structure can be identified. This has become an established approach for product identification [9,10,11]. In particular, TOF-MS spectrometry is highly accurate with a relative standard deviation of ~10^−5^. Obviously, this high accuracy further enhances the reliability of in situ product identification. 

In-situ product identification was applied to the phytophenol dimerization reactions of 13 phytophenol probes, including phenol, allylphenols, gingerols, capsaicin, tyrosine, esculetin, 3,6-dimethoxylcatechol, eugenol, isoeugenol, and coniferyl alcohol. Structurally, these phytophenols are similar but different, and thus they may construct a structure–function relationship (Figure 1 and Table 1). Through a systematical structure–function relationship analysis, the study tried to highlight the basic rules and stereochemical characters, which had not been described in the sporadic studies [12,13,14,15].

## 2. Results and Discussion

The first phytophenol probe, phenol (hydroxybenzene, **1**), was mixed with a solution of 2,2-diphenyl-1-picrylhydrazyl (DPPH^•^ radical) in methanol [16]. The product mixture was then analyzed in situ using UHPLC-ESI-Q-TOF-MS/MS. Thereafter, the possible dimeric products (**1a**, **1b,** or **1c**) were identified upon comparison with standard dimers. However, **1d** and **1e**, two dimeric products with the characteristic *m*/*z* 108 fragment (accurate *m*/*z* values of 108.0225 and 108.0238, respectively, Appendix A), were not detected. This agrees with a short report in 1968 [14]. Similar results were also found in the DPPH^•^-treated 4-allylphenol (**2**) reaction mixture, which produced magnolol (**2a**) and excluded 4,4′-diallyl-1-hydroxyldiphenyl ether (**2b**) (*m*/*z* 189). Interestingly, phytophenol 4-allylphenol (**2**) was cross-coupled with 2-allylphenol (**3**) to yield honokiol **3a** (Table 1 and Figure 1).

Figure 1 shows that the identified products (**1a**, **1b**, **1c**, **2a**, and **3a**) were constructed via C–C bond forming reactions. The three excluded ethers (**1d**, **1e**, and **2b**, Table 1) were linked via C–O bonding. This comparison indicates that C–C bonding plays a predominant role in the phytophenol dimerization reaction. This is referred to as the C–C bonding domination rule. This rule has also been supported in a previous study [15].

Handbook data suggest that the bond dissociation enthalpy (BDE) of the C–C bond is ~492 kJ/mol, whereas those of the C–O and O–O bonds are ~334 and ~131 kJ/mol, respectively. Thus, O–O bonding is essentially impossible because of the low BDE. However, C–O bonding can occasionally occur in the phytophenol dimerization reaction (e.g., C*_ortho_*–O dimer [13,17] and lignification [18]), whereas C–C bonding is dominant in the reaction [19,20]. 

From a resonance theory perspective, the phenolic -OH group donates a hydrogen atom and undergoes oxidation to form a ArO^•^ radical intermediate when interacting with free radicals. However, the unpaired electron of the O atom can resonate in the *ortho* or *para* position, but not in the *meta* position. Therefore, the *meta* position is inactive in the phytophenol dimerization reaction. Thus, the bonding site occurs at either the *ortho* or *para* position. If both the *ortho* and *para* positions are occupied, then the phytophenol dimerization reaction cannot occur at the *meta* position (e.g., 4, Table 1 and Figure 1). This is termed the *meta*-excluded rule.

The *meta*-exclusion rule is also applicable to long-branched phytophenols. For example, three long-branched phytophenol probes (**5**–**7**) were coupled with four *ortho*–*ortho* dimers (**5a**, **6a**, and **7a**, Table 1 and Figure 1). This means that the *meta*-excluded rule cannot be affected by long-branches despite the fact that the branch possesses great steric hindrance. Subsequently, long-branched tyrosine (**8**) occurred a dimerization reaction via the *ortho* bonding (**8a**, Figure 1). As tyrosine is the sole phenolic amino acid in protein, this finding partly explains the consistent location of protein nitration, lipidation, and cross-linking at the *ortho* position of tyrosine, [19] and the bonding site of echinatin–echinatin dimer [17]. 

All of these phytophenol probes (**1**–**8**) contain only one -OH° group. However, *ortho*-diOHs are also present in various phytophenols (e.g., coumestan wedelolactone [21]). Figure 1 (inset) shows that the *ortho*-diOH unit has two different positions, that is, the α- and β-position. The α-position can be regarded as the *ortho*-position of the next -OH group. In contrast, the β-position is actually the *para*-position to another -OH group. Thus, both the α- and β-position are activated. This is called the *ortho*-diOH co-activated rule. To confirm this rule, α-unoccupied euphorbetin (**9**) was treated with DPPH^•^. Table 1 and Figure 1 show that the reaction yields an α–α′ dimer (i.e., **9a** euphorbetin) or α–α‴ dimer (i.e., **9b** isoeuphorbetin). Under the same conditions, β-unoccupied 3,6-dimethoxylcatechol (**10**) yields a β–β′ dimer (**10a**). These results further imply that both the α- and β-positions were activated in the phytophenol dimerization reaction.

The last structural factor is the exocyclic C=C bond. Figure 1 shows that two phytophenol probes (**2** and **3**) have an exocyclic C=C bond. However, the two probes produce an acyclic dimer, suggesting that their phytophenol dimerization reactions fully comply with the *meta*-excluded and C–C bonding domination rules. Therefore, the exocyclic C=C bond is not involved in the reaction. A similar situation was observed in the dimerization reaction of eugenol (**11**).

Unlike eugenol (**11**), isoeugenol (**12**) affords dehydrodiisoeugenol (**12a**), a furan-fused dimer. The cyclization occurs at the original site of the exocyclic C=C bond (yellow in **12a,** Figure 1), indicating that the exocyclic C=C bond had already involved in the dimerization reaction. Exocyclic C=C involvement is attributed as the sole difference between **11** and **12**; i.e., the exocyclic C=C bond can conjugate with the phenyl ring in **12**, whereas that in **11** cannot. Consequently, if the exocyclic C=C bond conjugates with a phenyl ring, it will involve in the phytophenol dimerization reaction. Accordingly, conjugation with a phenyl ring is considered the first precondition for exocyclic C=C involvement. 

The first precondition was determined using quantum chemical calculations. Given that isoeugenol (**12**) donates a hydrogen atom via homolysis at the 4′-OH group, it is transformed into a 4′-O^•^ radical (***i***) (Figure 2). The calculation suggests that formula ***ii*** was found to be of the highest weight (18.35%). Thus, it may probably cross-couple with another radical intermediate ***v*** to construct the first covalent linkage (i.e., 3′-β in *vi*, Figure 2). The occurrence of formula ***v*** however is attributed to the situation that, (1) the exocyclic C=C bond is conjugative with the phenolic core; (2) the exocyclic C=C bond sits at the para-position of phenolic OH. If the -OH group was at the *meta*-position, then the unpaired electron of 12-ArO^•^ radical intermediate (***i***, Figure 2) would not move to the exocyclic site. This indicates that the location of the 4′-OH group at the *para* position was the second precondition.

Therefore, when the exocyclic C=C bond is conjugated with the phenyl ring and located at the *para*-position to the -OH group, it may involve in the phytophenol dimerization reaction. This is termed the exocyclic C=C involvement rule. In other words, once the exocyclic C=C bond adheres to these two preconditions, it can involve in the reaction. Exocyclic C=C involvement undoubtedly complicates the phytophenol dimerization reaction, especially the reaction pathway and stereochemistry.

Previously, there have been different opinions regarding the phytophenol dimerization reaction pathway with exocyclic C=C involvement. One study pointed out that, after the first radical coupling with ***v***, for example, ***vi*** mediated constructed the second covalent linkage, resulting in **12a** [22]. Our calculations refuted that there were evident electrical alternants in both the *semi*-quinone and 4′-OH group (***vii***, Figure 2). Such an alternating charge distribution could easily induce heterolysis rather than homolysis of the O–H bond at the 4′-OH group. In fact, its DPE (269.80 kcal/mol) was much lower than its BDE (392.06 kcal/mol), suggesting that the 4′-OH group would prefer to donate H^+^ and then transform into a 4′-O^−^ anion. Even though the 4′-OH group produces an ArO^•^ radical, there was no other radical for the coupling reaction. This is because the H atoms are attached to an sp^2^-hybridized C atom and thus, cannot homolyze to give another radical. Similarly, the H atoms in the semiquinone methide moiety of ***xix*** cannot homolyze to give another radical. In this case, the 4′-O^−^ anion accesses the α-C atom with +0.004 charge, to occur an intramolecular conjugate addition reaction (ICA). This is further supported by more biosynthetic studies reported in the literature [5,23,24,25]. After an ICA reaction and keto–enol tautomerism, intermediate ***xx*** was transformed into the final product (**12a**, Figure 2). Apparently, the production of **12a** is a result of stepwise pathway, including a simple radical coupling and a subsequent ICA reaction. The exploration of this mechanism has deepened the understanding of previous study [12].

To explore the stereochemistry, coniferyl alcohol (**13**) was used in this study. According to the discussion, the DPPH^•^-treated coniferyl alcohol is transferred into the **13** ArO^•^ radical intermediate, which is further transformed into a **13**βC^•^ intermediate via resonance (Figure 3). However, **13**βC^•^ is coupled into a quinone dimer and establishes a β, β′ covalent bond [2]. Undoubtedly, this is the basic phytophenol dimerization reaction. The Gaussian calculation showed that the phytophenol dimerization reaction always arrayed the βH,β′H atoms in *cis*- form in the optimized molecular model of the quinone dimer. However, the *cis*-βH,β′H atoms could exhibit two different directions, that is, βH,β′H upper and βH,β′H lower.

In the βH, β′H upper direction, the γ-OH attacked the α′-atom, while the γ′-OH attacked the α-atom via the ICA reaction. The two attacks were intramolecular, and thus they only allowed the *Si* surface attacks to prohibit the *Re* attacks. To effectively reduce the crowd, two huge groups the α-phenyl group and γ-CH_2_O group were arranged on the different side around the α-β σ bond, to show a preferential conformation, that is, **configuration 1** (Figure 3). In the configuration, αH and βH were at *trans*-form; while α′H and β′H were also at *trans*-form. It was marked as αRα′R βS β′S. This was actually (**−**) **13a**, a dimer with a furan-fused ring. Its diastereoisomer, (**−**) **13b** with *cis*-αH,βH configuration, could not be generated for *Re* prohibition.

Correspondingly, the βH,β′H lower direction enabled the formation of only (**+**) **13a**, in which there were *trans*-αH,βH and *trans*-α′H,β′H. Therefore, neither the upper direction nor the lower direction gave diastereoisomer (**−**) **13b**. This was evidenced by the chiral column-based HPLC-UV analysis in situ and UHPLC-ESI-Q-TOF-MS/MS analysis in situ. As seen in Appendix A and Figure 3, (−) epipinoresinol was absent in the product mixture of DPPH^•^-treated coniferyl alcohol, indicating that the exocyclic C=C-involved phytophenol dimerization reaction possesses high diastereoselectivity. In summary, the exocyclic C=C bond involved in the phytophenol dimerization reaction usually causes two adjacent chiral centers (e.g., α, β or α′, β′). However, the two cannot form a pair of diastereoisomers and are diastereoselective.

The findings regarding the diastereoselectivity in the exocyclic C=C-involved phytophenol dimerization reaction has further deepened the understanding of stereochemistry in lignin dimerization, a basic lignin biosynthesis reaction. Previously, biochemists have investigated the stereoselectivity of lignin dimerization. However, the stereoselectivity is enantioselectivity rather than diastereoselectivity [2]. The enantioselectivity was determined from the enzymatic specification, which is an issue in biochemistry [26].

In comparison, the present diastereoselectivity finding can be ultimately regarded as a chemical issue, because it was also observed in the DPPH^•^-treated chemical system. Therefore, this finding may also benefit chemists. For example, List and co-workers used asymmetric catalysis to control conjugate addition. It was reported that they could not achieve a high diastereoselectivity, because the *Re* attack was inevitable; thus, the products always contained a pair of diastereoisomers in two adjacent chiral centers. The great difference apparently outperforms the diastereoselective feature of the exocyclic C=C-involved phytophenol dimerization reaction. Meanwhile, it offers organic chemists the advice that designing an intramolecular reaction may be a more effective strategy than asymmetric catalysis to control conjugate addition.

Similar diastereoselective products were also found in HRP/H_2_O_2_-treated coniferyl alcohol (Appendix A and Figure 3). The similarity between the DPPH^•^-treated system and the HRP/H_2_O_2_-treated system suggests that HRP-catalyzed lignin dimerization is essentially a radical-mediated process. This proposal was further supported by previous reports that HRP-mediated reactive oxygen species (ROS) fulfills phenolic lignin biosynthesis. However, ROS are mainly free radicals; therefore, HRP/H_2_O_2_ mediated phenolic (especially phenolic lignin) biosynthesis is essentially an exocyclic C=C-involved phytophenol dimerization reaction [2,27]. Therefore, enzymatic lignin dimerization was also deduced to be diastereoselective. This could be the reason why the natural 3,7-dioxabicyclo [3.3.0] octane lignin skeleton usually arranges the *trans*-αH,βH (or *trans*-α′H,β′H) at the a furan-fused ring (Appendix A).

In this study, two phytophenol probes were used to discuss the stepwise pathway and diastereoselectivity of the exocyclic C=C-involved phytophenol dimerization reaction. It can be summarized that, when an exocyclic C=C bond is conjugated with the phenolic core and has a *para*-OH group, it may be involved in the phytophenol dimerization reaction. The exocyclic C=C-involved phytophenol dimerization reaction proceeds via a stepwise pathway. The first step is a basic radical coupling reaction, whereas the subsequent step is an ICA reaction. After two steps, a dimer with a furan-fused ring may be generated. Despite the furan-fused ring bearing two adjacent chiral atoms, these two atoms cannot form a pair of diastereoisomers because of intramolecular prohibitions. The diastereoselectivity was associated with the stepwise pathway. The exocyclic C=C involvement rule, along with the three basic rules, can explain and even predict the products in chemical and biological fields, especially lignin biosynthesis and organic chemistry. Other relevant fields may also include natural product chemistry, synthetic chemistry, enzyme chemistry, neurochemistry, electrochemistry, food chemistry, biochemistry, metabolic chemistry, photochemistry, and environmental chemistry (Appendix A). These great implications can be attributed to the fact that: (1) “Phytophenols” cover a very wide range of compounds, such as stilbene, phenolic acid, phenolic amino acids, lignin, phenylpropanoid, volatile phenols, and coumarin, and (2) in addition to various phytophenols, free radicals, the initiator of the phytophenol dimerization reaction, are ubiquitous in plants and nature.

## 3. Conclusions

In conclusion, the phytophenol dimerization reaction is governed by four basic rules: *meta*-excluded, C-C bonding domination, *ortho*-diOH coactivation, and exocyclic C=C involvement. Exocyclic C=C involvement requires conjugation with the aromatic ring and location at the *para* position to the phenolic -OH group. The exocyclic C=C-involved phytophenol dimerization reaction proceeds via a stepwise pathway, which is a simple radical coupling reaction and a subsequent intramolecular conjugate addition. These reactions usually yield a furan-fused ring with two adjacent chiral atoms. However, the two adjacent chiral atoms are highly diastereoselective owing to intramolecular prohibitions. These findings can explain why oxidase-catalyzed lignin radical-based cyclization cannot produce a pair of diastereomers at two adjacent chiral atoms.

## Figures and Tables

**Figure 1 molecules-27-04842-f001:**
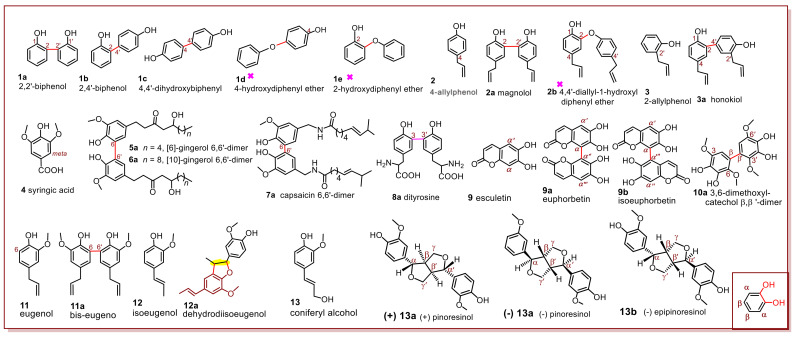
Structures of some phytophenols mentioned in Table 1. (The absolute stereo-configuration 12a has not been fully identified by chemists nowadays. Inset shows numbering of the *ortho*-diOH unit. The red “X” denotes the excluded product).

**Figure 2 molecules-27-04842-f002:**
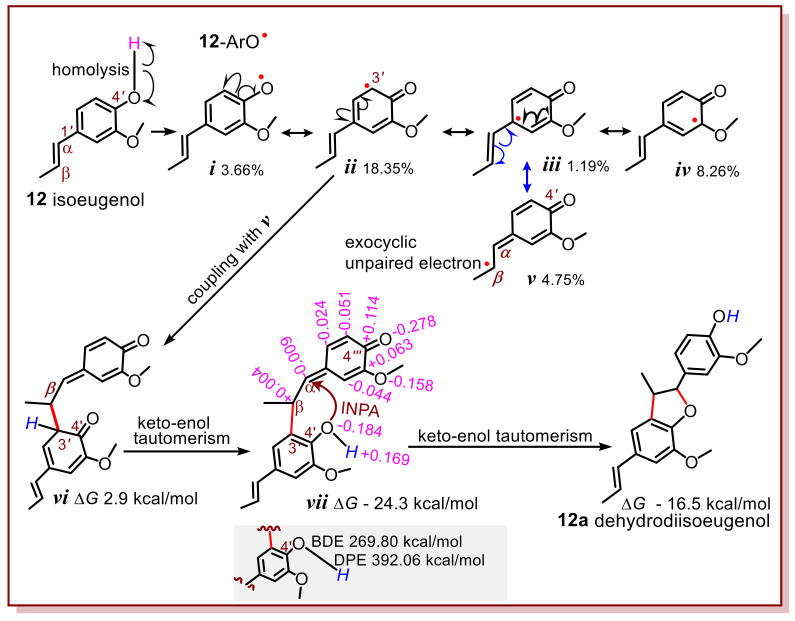
The possible process from isoeugenol (**12**) to dehydrodiisoeugenol (**12a**) based on computational chemistry. Documental **12a** has no absolute configuration. The pink numbers represent the Hirshfeld charges of atoms. The percentages under ***i***–***v*** refer to the weights of the resonance formula in the natural resonance theory (NRT) analysis. Δ*G* means the change in Gibbs free energy. ICA, intramolecular conjugate addition; BDE, bond disassociation enthalpy; DPE, deprotonation enthalpy. The computational details were shown in Appendix A.

**Figure 3 molecules-27-04842-f003:**
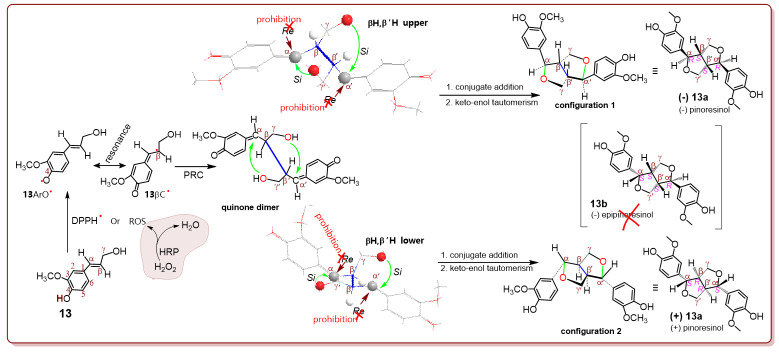
The proposed process and stereochemistry of the conversion of **13** into (−) **13a** mediated by horseradish peroxidase (HRP)/H_2_O_2_ or DPPH^•^ radical. The blue line indicates the β,β′ covalent bond stemming from the basic phytophenol dimerization reaction.

**Table 1 molecules-27-04842-t001:** Thirteen phytophenol probes, 15 identified dimeric products, and five excluded dimeric products.

Phytophenol Probe	Identified Product	Excluded Product
phenol (**1**)	**1a**, **1b**	**1c**, **1e**, **1d**
4-allylphenol (**2**)	**2a**	**2b**
4-allylphenol (**2**) *plus* 2-allylphenol (**3**)	**3a**	
syringic acid (**4**)	No product	
[6]-gingerol (**5**)	**5a**	
[10]-gingerol (**6**)	**6a**	
capsaicin (**7**)	**7a**	
tyrosine (**8**)	**8a**	
esculetin (**9**)	**9a** (or **9b**)	
3,6-dimethoxylcatechol (**10**)	**10a**	
eugenol (**11**)	**11a**	
isoeugenol (**12**)	**12a**	
coniferyl alcohol (**13**)	(**+**)**13a**, (**−**) **13a**	**13 b**

Note: The dimeric product was obtained via the DPPH^•^ -initiated phytophenol dimerization reaction performed in methanol. The identification or exclusion of products was based on the high-resolution MS spectra obtained from UHPLC-ESI-Q-TOF-MS/MS in situ analysis with the aid of newly synthesized and authentic standards (Appendix A). Dimeric product 9 was identified as either **9a** or **9b**. However, these uncertainties cannot hinder the discussion regarding basic rules. The so-called full product identification was a merely qualitative result, because the quantitative analysis was a of UHPLC-ESI-Q-TOF-MS/MS.

## Data Availability

The data presented in this study are available in this article.

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
