# Peer review of "Phytophenol Dimerization Reaction: From Basic Rules to Diastereoselectivity and Beyond"

_molecules, 2022, doi:10.3390/molecules27154842_

Round 1
Reviewer 1 Report
This manuscript describes the radical-mediated dimerization reaction of 13 phytophenols and in situ product identification using UHPLC-ESI-Q-TOF-MS/MS. The authors found out four basic rules governing phytophenol dimerization reaction: meta-exclusion, C-C bonding domination, ortho-di-OH coactivation, and exocyclic C=C bond involvement. This study was systematically described and presented reasonable conclusions from the comparison of MS data between reaction products and authentic samples. However, it is recommended to provide supporting evidence by isolating the dimerization reaction products of at least one of the 13 phytophenols and confirming their structures with other spectroscopic data. Also, please, change disassociation(p 2 line 88) to dissociation. This paper is recommended for publication in Molecules.
Author Response
Please kindly see the attached pdf file.

Reviewer 2 Report
Review comments to the author
Title: ''Phytophenol Dimerization Reaction: from Basic Rules to Diastereoselectivity and Beyond''.
Manuscript ID: molecules-1829639.
Introduction:
1- Page 1, Line 38: The citations [2], [3] should be written as [2, 3]. Please, apply this concept for all coming citations.
2- Page 2, Line 54: Add one space before the citation [6, 7]. Please, apply this concept for all coming citations.
Results and discussion:
1- Page 4, Line 143, Table 1: The title of the first column should be written as ''Phytophenol probe''.
2- Page 5, Line 192: Add one space after the citation [19-21].
3- Page 5, Line 196: Add one space before the citation [11].
4- Page 5, Line 166-177: This section should be reinforced by relevant citations.
Abbreviations:
- List of abbreviations should be inserted by the end of the manuscript before references.
References:
1- Page 8, Line 338: The species names ''Bacillus megaterium'' and ''Curvularia lunata'' should be written in italic font.
Supplementary Materials:
Chemicals and reagents
1- It is necessary to add a new section containing all the chemicals and reagents used and their sources (Company name, city and country).
